# SuperEncoder: Towards Iteration-Free Approximate Quantum State Preparation

## Abstract

Numerous quantum algorithms operate under the assumption that classical data has already been converted into quantum states, a process termed Quantum State Preparation (QSP). However, achieving precise QSP requires a circuit depth that scales exponentially with the number of qubits, making it a substantial obstacle in harnessing quantum advantage. Recent research suggests using a Parameterized Quantum Circuit (PQC) to approximate a target state, offering a more scalable solution with reduced circuit depth compared to precise QSP. Despite this, the need for iterative updates of circuit parameters results in a lengthy runtime, limiting its practical application. To overcome this challenge, we introduce SuperEncoder, a pre-trained classical neural network model designed to directly estimate the parameters of a PQC for any given quantum state. By eliminating the need for iterative parameter tuning, SuperEncoder represents a pioneering step towards iteration-free approximate QSP.

## 1 Introduction

Quantum Computing (QC) leverages quantum mechanics principles to address classically intractable problems [47, 36]. Various quantum algorithms have been developed, encompassing quantum-enhanced linear algebra [15, 48, 45], Quantum Machine Learning (QML) [26, 19, 1, 33, 50, 3], quantum-enhanced partial differential equation solvers [31, 13], etc. A notable caveat is that those algorithms assume that classical data has been efficiently loaded into a specific quantum state, a process known as Quantum State Preparation (QSP).

However, the realization of QSP presents significant challenges. Ideally, we expect each element of the classical data to be precisely transformed into an amplitude of the corresponding quantum state. This precise QSP is also known as Amplitude Encoding (AE). However, a critical yet unresolved problem of AE is that the required circuit depth grows exponentially with respect to the number of qubits [34, 41, 29, 46, 49]. Extensive efforts have been made to alleviate this issue, but they fail to address it fundamentally. For example, while some methods introduce ancillary qubits for shallower circuit [57, 56, 2], they may encounter an exponential number of ancillary qubits. Other methods aim at preparing *special* quantum states with lower circuit depth, being only effective for either sparse states [12, 32] or states with some special distributions [14, 17]. To summarize, realizing AE for *arbitrary* quantum states still remains *non-scalable* due to its exponential resource requirement with respect to the number of qubits. Moreover, in the Noisy Intermediate-Scale Quantum (NISQ) era [42], hardware has limited qubit lifetimes and confronts a high risk of decoherence errors when executing deep circuits, further exacerbating the problem of AE.

In fact, precise QSP is unrealistic in the present NISQ era due to the inherent errors of quantum devices. Hence, iteration-based Approximate Amplitude Encoding (AAE) emerges as a promising technique [59, 35, 52]. Specifically, AAE constructs a quantum circuit with tunable parameters, then

it iteratively updates the parameters to approximate a target quantum state. Since the updating of parameters can be guided by states obtained from noisy devices, AAE is robust to noises, becoming especially suitable for NISQ applications. More importantly, AAE has been shown to have shallow circuit depth [35, 52], making it more scalable than AE.

Unfortunately, AAE possesses a drawback that significantly undermines its potential advantages — the lengthy runtime stemming from iterative optimizations of parameters. For example, when a Quantum Neural Network (QNN) [3] is trained and deployed, the runtime of AAE dominates the inference time as we demonstrated in Fig. 1. Since loading classical data into quantum states becomes the bottleneck, the potential advantage of QNN diminishes no matter how efficient the computations are done on quantum devices.

Compared to AAE, AE employs a pre-defined arithmetic decomposition procedure to construct a circuit, thereby becoming much *faster* than AAE at runtime. Therefore, it is natural to ask: can we realize both *fast* and *scalable* methods for *arbitrary* QSP? This is precisely the question we tackle in this paper. Overall, we present three major contributions.

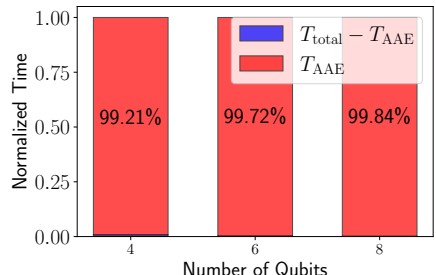

Figure 1: Breakdown of normalized runtime for QNN inference. Original data are listed in Table 1.

- Given a Parameterized Quantum Circuit (PQC) $U(\boldsymbol{\theta})$ that approximates a target quantum state, with $\boldsymbol{\theta}$ the parameter vector. We show that there exists a *deterministic* transformation $f$ that could map an arbitrary state $|\boldsymbol{d}\rangle$ to its corresponding parameters $\boldsymbol{\theta}$. Consequently, the parameters can be designated by $f$ without time-intensive iterations.
- We show that the mapping $f$ is *learnable* by utilizing a classical neural network model, which we term as SuperEncoder. With SuperEncoder, you can have your cake and eat it too, i.e., simultaneously realizing *fast* and *scalable* QSP. We develop a prototype model and shed light on insights into its training methodology.
- We verify the effectiveness of SuperEncoder on both synthetic dataset and representative downstream tasks, paving the way toward iteration-free approximate quantum state preparation.

## 2 Preliminaries

In this section, we commence with some basic concepts about quantum computing [36], and then proceed to a brief retrospect of existing QSP methods.

### 2.1 Quantum Computation

We use Dirac notation throughout this paper. A *pure quantum state* is defined by a vector $|\cdot\rangle$ named 'ket', with the unit length. A state can be written as $|\psi\rangle = \sum_{j=1}^{N} \alpha_j |j\rangle$ with $\sum_j |\alpha_j|^2 = 1$, where $|j\rangle$ denotes a computational basis state and $N$ represents the dimension of the complex vector space. *Density operators* describe more general quantum states. Given a mixture of $m$ pure states $\{|\psi_i\rangle\}_{i=1}^{m}$ with probabilities $p_i$ and $\sum_i^m p_i = 1$, the density operator $\rho$ denotes the *mixed state* as $\rho = \sum_{i=1}^{m} p_i |\psi_i\rangle\langle\psi_i|$ with $\text{Tr}(\rho) = 1$, where $\langle\cdot|$ refers to the conjugate transpose of $|\cdot\rangle$. Generally, we use the term *fidelity* to describe the similarity between an erroneous quantum state and its corresponding correct state.

The fundamental unit of quantum computation is the quantum bit, or *qubit*. A qubit's state can be expressed as $\psi = \alpha|0\rangle + \beta|1\rangle$. Given $n$ qubits, the state is generalized to $|\psi\rangle = \sum_{j}^{2^n} |j\rangle$, where $|j\rangle = |j_1 j_2 \cdots j_n\rangle$ with $j_k$ the state of $k$th qubit in computational basis, and $j = \sum_{k=1}^{n} 2^{n-k} j_k$. Applying *quantum operations* evolves a system from one state to another. Generally, these operations can be categorized into quantum gates and measurements. Typical single-qubit gates include the Pauli gates $X \equiv \left[\begin{smallmatrix} 0 & 1 \\ 1 & 0 \end{smallmatrix}\right]$, $Y \equiv \left[\begin{smallmatrix} 0 & -i \\ i & 0 \end{smallmatrix}\right]$, $Z \equiv \left[\begin{smallmatrix} 1 & 0 \\ 0 & -1 \end{smallmatrix}\right]$. These gates have associated rotation operations $R_P(\theta) \equiv e^{-i\theta P/2}$, where $\theta$ is the rotation angle and $P \in \{X, Y, Z\}$[1]. Muti-qubit operations create

---

[1]In this paper, $R_z$, $R_y$ are equivalent to $R_Z$, $R_Y$.

*entanglement* between qubits, allowing one qubit to interfere with others. In this work, we focus on the controlled-NOT (CNOT) gate, with the mathematical form of $\text{CNOT} \equiv |0\rangle\langle 0| \otimes \mathbf{I}_2 + |1\rangle\langle 1| \otimes X$. Quantum measurements extract classical information from quantum states, which is described by a collection $\{M_m\}$ with $\sum_m M_m^\dagger M_m = \mathbf{I}$. Here, $m$ refers to the measurement outcomes that may occur in the experiment, with a probability of $p(m) = \langle\psi|M_m^\dagger M_m|\psi\rangle$. The post-measurement state of the system becomes $M_m|\psi\rangle/p(m)$.

A *quantum circuit* is the graphical representation of a series of quantum operations, which can be mathematically represented by a unitary matrix $U$. In the NISQ era, PQC plays an important role as it underpins variational quantum algorithms [11, 39]. Typical PQC has the form of $U(\boldsymbol{\theta}) = \prod_i U_i(\theta_i)V_i$, where $\boldsymbol{\theta}$ is its parameter vector, $U_i(\theta_i) = e^{-i\theta_i P_i/2}$ with $P_i$ denoting a Pauli gate, and $V_i$ denotes a fixed gate such as CNOT. For example, a PQC composed of $R_y$ gates and CNOT gates is depicted in Fig. 2.

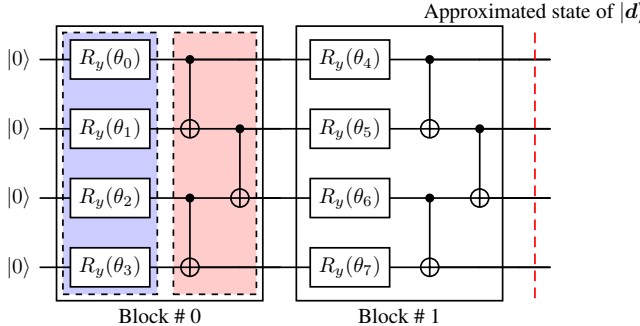

Figure 2: An example PQC with two blocks, with each block consisting of a rotation layer (filled blue) plus an entangler layer (filled red).

## 2.2 Quantum State Preparation

Successful execution of many quantum algorithms requires an initial step of loading classical data into a quantum state [5, 15], a process known as *quantum state preparation*. This procedure involves implementing a quantum circuit to evolve a system to a designated state. Here, we focus on *amplitude encoding* and formalize its procedure as follows. Let $\boldsymbol{d}$ be a real-valued $N$-dimensional classical vector, AE encodes $\boldsymbol{d}$ into the amplitudes of an $n$-qubit quantum state $|\boldsymbol{d}\rangle$, where $N = 2^n$. More specifically, the data quantum state is represented by $|\boldsymbol{d}\rangle = \sum_{j=0}^{N-1} d_j|j\rangle$, where $d_j$ denotes the $j$th element of the vector $\boldsymbol{d}$, and $|j\rangle$ refers to a computational basis state. The main objective is to generate a quantum circuit $U$ that initializes an $n$-qubit system by $U|0\rangle^{\otimes n} = \sum_{j=0}^{N-1} \alpha_j|j\rangle$, whose amplitudes $\{\alpha_j\}$ are equal to $\{d_j\}$. It is widely recognized that constructing such a circuit generally necessitates a circuit depth that scales exponentially with $n$ [34, 41]. This property makes AE impractical in current NISQ era, as decoherence errors [23] can severely dampen the effectiveness of AE as the number of qubits increases [52].

In response to the inherent noisy nature of current devices, *approximate amplitude encoding* has emerged as a promising technique [59, 35, 52]. Specifically, AAE utilizes a PQC (a.k.a. ansatz) to approximate the target quantum state by iteratively updating the parameters of circuit, following a similar procedure of other variational quantum algorithms [39, 11]. AAE has been shown to be more advantageous for NISQ devices due to its ability to mitigate coherent errors through flexible adjustment of circuit parameters, coupled with its lower circuit depth [52]. We denote an ansatz as $U(\boldsymbol{\theta})$, where $\boldsymbol{\theta}$ refers to a vector of tunable parameters for optimizations. A typical ansatz consists of several blocks of operations with the same structure. For example, a two-block ansatz with 4 qubits is shown in Fig. 2, where the rotation layer is composed of single-qubit rotational gates $R_y(\theta_r) = e^{-i\theta_r Y/2}$, and the entangler layer comprises CNOT gates. Note that the entangler layer is configurable and hardware-native, which means that we can apply CNOT gates to physically adjacent qubits, thereby eliminating the necessity of additional SWAP gates to overcome the topological constraints [27]. This type of PQC is also known as *hardware-efficient ansatz* [20], being widely adopted in previous studies of AAE [59, 35, 52].

## 3 SuperEncoder

### 3.1 Motivation

Although AAE can potentially realize high fidelity QSP with $O(\text{poly}(n))$ circuit depth [35] with $n$ the number of qubits, it requires repetitive *online* tuning of parameters to approximate the target state, which may result in an excessively long runtime that undermines its feasibility. Specifically, we could consider a simple application scenario in QML. The workflow with AAE is depicted in Fig. 3a. During the inference stage, we must iteratively update the parameters of the AAE ansatz for each input classical data vector, which may greatly dampen the performance. To quantify this impact, we measure the runtime of AAE-based data loading and the total runtime of model inference. As one can observe from Table 1, AAE dominates the runtime, thereby becoming the performance bottleneck.

| $n$ | $T_{\text{AAE}}$ (s) | $T_{\text{total}} - T_{\text{AAE}}$ (s) |
|---|---|---|
| 4 | **5.0086** | 0.0397 |
| 6 | **20.1810** | 0.0573 |
| 8 | **59.4193** | 0.0978 |

Table 1: **Performance overhead of AAE**. We break down the averaged inference runtime per sample from the MNIST dataset. $T_{\text{AAE}}$ denotes time spent on loading classical data into quantum state using AAE, and $T_{\text{total}}$ refers to total runtime.

The necessity of time-intensive iterations is grounded in the following assumption — Given an arbitrary quantum state $|\psi\rangle$, there *does not* exist a deterministic transformation $f : |\psi\rangle \rightarrow \boldsymbol{\theta}$, where $\boldsymbol{\theta}$ refers to the vector of parameters enabling a PQC to prepare an approximated state of $|\psi\rangle$. This assumption seems intuitively correct given the randomness of target states. However, we argue that a universal mapping $f$ exists for any arbitrary data state $|\psi\rangle$. Taking a little thought of AE, we see that it implies the following conclusion: given an arbitrary state $|\psi\rangle$, there exists an universal arithmetic decomposition procedure $g : |\psi\rangle \rightarrow U$ satisfying $U|0\rangle = |\psi\rangle$. Inspired by this deterministic transformation, it is natural to ask: is there an universal transformation $g' : |\psi\rangle \rightarrow U'$ satisfying $E(U'|0\rangle, |\psi\rangle) \leq \epsilon$? Here $E$ denotes the deviation between the prepared state by a circuit $U'$ and the target state, and $\epsilon$ refers to certain acceptable error threshold. Since the structure of PQC in AAE is the same for any target state, $U'$ is determined by $\boldsymbol{\theta}$. Then, the problem is reduced to exploring the existence of $f : |\psi\rangle \rightarrow \boldsymbol{\theta}$. Should $f$ exist, the overhead of online iterations could be eliminated, resulting in a novel QSP method being both fast and scalable.

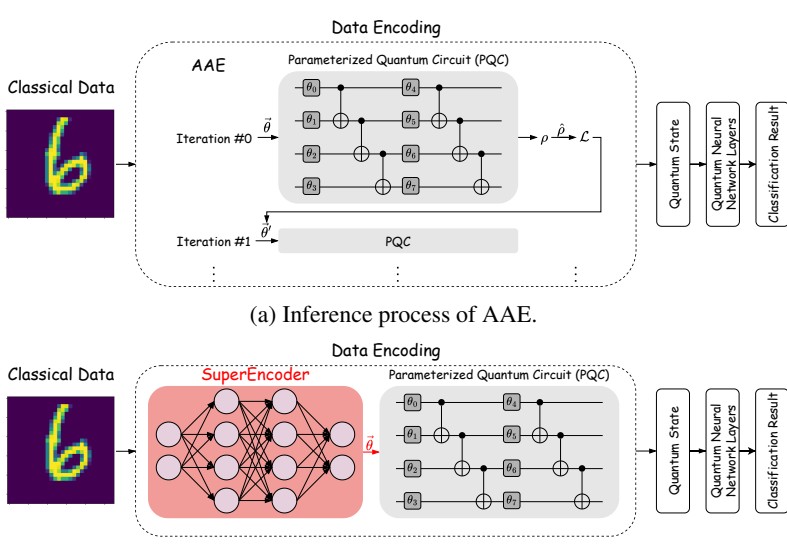

(a) Inference process of AAE.

(b) Inference process of SuperEncoder.

Figure 3: Comparison between AAE and SuperEncoder.

## 3.2 Design Methodology

Let $|\psi\rangle$ be the target state, and $U(\boldsymbol{\theta})$ be the PQC used in AAE with $\boldsymbol{\theta}$ the optimized parameters. Our goal is to develop a model, termed SuperEncoder, to approximate the mapping $f : |\psi\rangle \rightarrow \boldsymbol{\theta}$. Referring back to the scenario in QML, the workflow with SuperEncoder becomes iteration-free, as depicted in Fig. 3b.

Since neural networks could be used to approximate any continuous function [6], a natural solution is to use a neural network to approximate $\hat{f}$. Specifically, we adopt a Multi-Layer Perceptron (MLP) as the backbone model for approximating $f$. However, training this model is nontrivial. Particularly, we find it challenging to design a proper loss function. In the remainder of this section, we explore three different designs and analyze their performance.

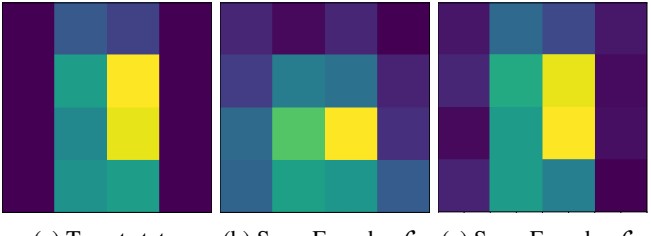

(a) Target state. (b) SuperEncoder-$\mathcal{L}_1$ (c) SuperEncoder-$\mathcal{L}_3$

Figure 4: Virtualization of states generated by SuperEncoder trained with different loss functions. $\mathcal{L}_2$ is omitted as it produces very similar results to $\mathcal{L}_3$.

The first and most straightforward method is *parameter-oriented* training — setting the loss function $\mathcal{L}_1$ as the MSE between the target parameters $\boldsymbol{\theta}$ from AAE and the output parameters $\hat{\boldsymbol{\theta}}$ from SuperEncoder. To evaluate the performance of $\mathcal{L}_1$, we train a SuperEncoder using MNIST dataset, and test if it could load a test digit image into a quantum state with high fidelity. All images are downsampled and normalized into 4-qubit states for quick evaluation.

Unfortunately, results in Table 2 show that $\mathcal{L}_1$ achieves poor performance. The average fidelity of prepared quantum states is only 0.6208. As demonstrated in Fig. 4, $\mathcal{L}_1$ generates a state that losses the patterns of the original state. Additionally, utilizing $\mathcal{L}_1$ implies that we need to first generate target parameters using AAE, of which the long runtime hinders pre-training on larger datasets. Consequently, required is a more effective loss function design without involving AAE.

| $\mathcal{L}_1$ | $\mathcal{L}_2$ | $\mathcal{L}_3$ |
|---|---|---|
| 0.6208 | 0.9873 | 0.9908 |

Table 2: Fidelity comparison between SuperEncoders trained with different loss functions.

To address this challenge, we propose a *state-oriented* training methodology, which employs quantum states as targets to guide optimizations. Specifically, we may apply $\hat{\boldsymbol{\theta}}$ to the circuit and execute it to obtain the prepared state $\hat{\psi}$. Then it is possible to calculate the difference between $\hat{\psi}$ and $\psi$ as the loss to optimize SuperEncoder. In contrast to parameter-oriented training, this approach applies to larger datasets as it decouples the training procedure from AAE. We utilize two different state-oriented metrics, the first being the MSE between $\hat{\psi}$ and $\psi$, denoted as $\mathcal{L}_2$, and the second is the *fidelity* of $\hat{\psi}$ relative to $\psi$, expressed as $\mathcal{L}_3 = 1 - |\langle\hat{\psi}|\psi\rangle|^2$ [25]. Results in Table 2 show that $\mathcal{L}_2$ and $\mathcal{L}_3$ achieve remarkably higher fidelity than $\mathcal{L}_1$. Besides, we observe that $\mathcal{L}_3$ prepares a state very similar to the target one (Fig. 4), verifying that state-oriented training is more effective than parameter-oriented training.

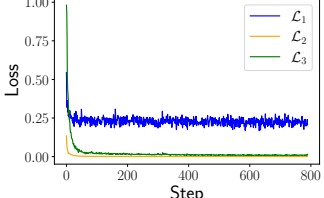

Figure 5: Convergence of different loss functions.

**Landscape Analysis**. To understand the efficacy of these loss functions, we further analyze their landscapes following previous studies [28, 40, 18]. To gain insight from the landscape, we plot Fig. 6 using the same scale and color gradients [18]. Compared to state-oriented losses ($\mathcal{L}_2$ and $\mathcal{L}_3$), $\mathcal{L}_1$ has a largely flat landscape with non-decreasing minima, thus the model struggles to explore a viable path towards a lower loss value, a similar pattern can also be observed in Fig. 5. In contrast, $\mathcal{L}_2$

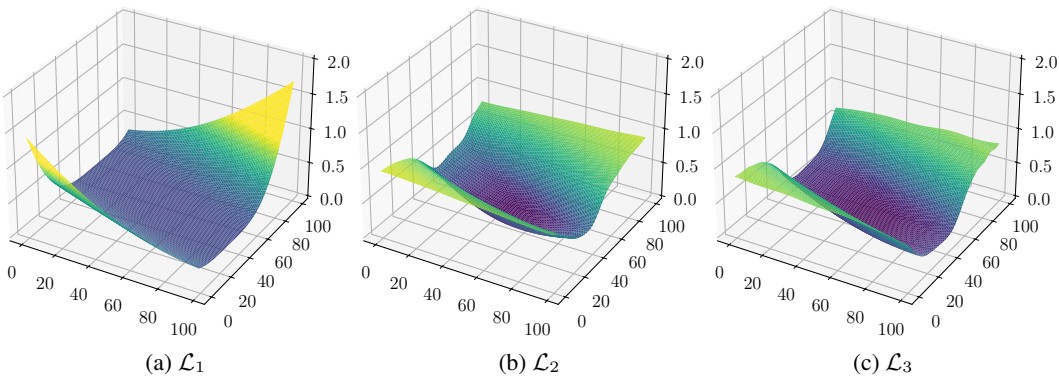

(a) $\mathcal{L}_1$      (b) $\mathcal{L}_2$      (c) $\mathcal{L}_3$

Figure 6: Landscape virtualization of different loss functions.

and $\mathcal{L}_3$ have much lower minima and successfully converge to smaller loss values. Furthermore, we observe from Fig. 6 that $\mathcal{L}_3$ has a wider minima than $\mathcal{L}_2$, which may indicate a better generalization capability [40].

**Gradient Analysis**. Based on the landscape analysis, we adopt $\mathcal{L}_3$ as the loss function to train SuperEncoder. We note that $\mathcal{L}_3$ can be written as $1 - \langle\psi|\hat{\psi}\rangle\langle\hat{\psi}|\psi\rangle$. If $\hat{\rho}$ is a pure state, it is equivalent to $|\hat{\psi}\rangle\langle\hat{\psi}|$. Then $\mathcal{L}_3$ is given by $\mathcal{L}_3 = 1 - \langle\psi|\hat{\rho}|\psi\rangle$.

This re-formalization is important as only the mixed state $\hat{\rho}$ could be obtained in noisy environments. Suppose an $n$-qubit circuit is parameterized by $m$ parameters $\hat{\boldsymbol{\theta}} = [\hat{\theta}_1, \ldots, \hat{\theta}_k, \ldots, \hat{\theta}_m]$. Let $\mathbf{W}$ be the weight matrix of MLP, with $k, l$ the element indices. We analyze the gradient of $\mathcal{L}_3$ w.r.t. $W_{k,l}$ to showcase its feasibility in different quantum computing environments.

$$\nabla_{W_{k,l}}\mathcal{L}_3 = \frac{\partial\mathcal{L}_3}{\partial W_{k,l}} = -\langle\psi|\frac{\partial\hat{\rho}}{\partial W_{k,l}}|\psi\rangle$$

$$= -\langle\psi| \begin{bmatrix} \sum_{j=1}^m \frac{\partial\hat{\rho}_{1,1}}{\partial\theta_j}\frac{\partial\theta_j}{\partial W_{k,l}} & \cdots & \sum_{j=1}^m \frac{\partial\hat{\rho}_{1,N}}{\partial\theta_j}\frac{\partial\theta_j}{\partial W_{k,l}} \\ \vdots & \ddots & \vdots \\ \sum_{j=1}^m \frac{\partial\hat{\rho}_{N,1}}{\partial\theta_j}\frac{\partial\theta_j}{\partial W_{k,l}} & \cdots & \sum_{j=1}^m \frac{\partial\hat{\rho}_{N,N}}{\partial\theta_j}\frac{\partial\theta_j}{\partial W_{k,l}} \end{bmatrix} |\psi\rangle, \tag{1}$$

The calculation of $\frac{\partial\theta_j}{\partial W_{k,l}}$ can be easily done on classical devices using backpropagation supported by automatic differentiation frameworks. Therefore, we only focus on $\frac{\partial\hat{\rho}_{i,j}}{\partial\theta_k}$. In a simulation environment, the calculation of $\hat{\rho}$ is conducted via noisy quantum circuit simulation, which is essentially a series of tensor operations on state vectors. Therefore, the calculation of $\frac{\partial\hat{\rho}_{i,j}}{\partial\theta_k}$ is compatible with backpropagation. The situation on real devices becomes more complicated. On real devices, the mixed state $\hat{\rho}$ is reconstructed through *quantum tomography* [7] based on classical shadow [55, 16]. Here, for notion simplicity, we denote the process of classical shadow as a transformation $\mathcal{S}$, and denote the measurement expectations of the ansatz as $U(\hat{\boldsymbol{\theta}})$. Thus the reconstructed density matrix is given by $\hat{\rho} = \mathcal{S}(U(\hat{\boldsymbol{\theta}}))$. Then the gradient of $\hat{\rho}_{i,j}$ with respect to $\hat{\theta}_k$ is $\sum_u \frac{\partial\hat{\rho}_{i,j}}{\partial U(\hat{\boldsymbol{\theta}})}\frac{\partial U(\hat{\boldsymbol{\theta}})}{\partial\hat{\theta}_k}$. Here $\frac{\partial\hat{\rho}_{i,j}}{\partial U(\hat{\boldsymbol{\theta}})}$ can be efficiently calculated on classical devices using backpropagation, as $\mathcal{S}$ operates on expectation values on classical devices. However, $U(\hat{\boldsymbol{\theta}})$ involves state evolution on quantum devices, where back-propagation is impossible due to the No-Cloning theorem [36]. Fortunately, it is possible to utilize the *parameter shift* rule [8, 4, 53] to calculate $\frac{\partial U(\hat{\boldsymbol{\theta}})}{\partial\theta_k}$. In this way, the gradients of the circuit function $U$ with respect to $\theta_j$ are $\frac{\partial U(\hat{\boldsymbol{\theta}})}{\partial\theta_k} = \frac{1}{2}\left(U(\theta_+) - U(\theta_-)\right)$, where $\theta_+ = [\theta_1, \ldots, \theta_k + \frac{\pi}{2}, \ldots, \theta_m], \theta_- = [\theta_1, \ldots, \theta_k - \frac{\pi}{2}, \ldots, \theta_m]$. To summarize, training SuperEncoder with $\mathcal{L}_3$ is theoretically feasible on both simulators and real devices.

## 4 Numerical Results

### 4.1 Experiment Setup

**Datasets.** To train a SuperEncoder for arbitrary quantum states, we need a dataset comprising a wide range of quantum states with different distributions. To our knowledge, there is no dataset dedicated for this special purpose. A natural solution is to use readily available datasets from classical machine learning domains (e.g., ImageNet [9], Places [58], SQuAD [44]) by normalizing them to quantum states. However, QSP is essential in various application scenarios besides QML. The classical data to be loaded may not only contain natural images or languages but also contain arbitrary data (e.g., in HHL algorithm [15]). Therefore, we construct a training dataset adapted from FractalDB-60 [21] with 60k samples, a formula-driven dataset originally designed for computer vision without any natural images. We also construct a separate dataset to test the performance of QSP, which consists of data sampled from different statistical distributions, including uniform, normal, log-normal, exponential, and Dirichlet distributions, with 3000 samples per distribution. Hereafter we refer this dataset as the *synthetic dataset*.

**Platforms.** We implement SuperEncoder using PennyLane [34], PyTorch [37] and Qiskit [43]. Simulations are done on a Ubuntu server with 768 GB memory, two 32-core Intel(R) Xeon(R) Silver 4216 CPU with 2.10 GHz, and 2 NVIDIA A-100 GPUs. IBM quantum cloud platform[2] is adopted to evaluate the performance on real quantum devices.

**Metrics.** We evaluate SuperEncoder and compare it to AE and AAE in terms of runtime, scalability, and fidelity. *Runtime* refers to how long it takes to prepare a quantum state. *Scalability* refers to how the circuit depth grows with the number of qubits. *Fidelity* evaluates the similarity between prepared quantum states and target quantum states. Specifically, the fidelity for two mixed states given by density matrices $\rho$ and $\hat{\rho}$ is defined as $F(\rho, \hat{\rho}) = \mathrm{Tr} \left( \sqrt{\sqrt{\rho} \hat{\rho} \sqrt{\rho}} \right)^2 \in [0, 1]$. A larger $F$ indicates a better fidelity.

**Implementation.** We implement SuperEncoder using an MLP consisting of two hidden layers. The dimensions of input and output layers are respectively set to $2^n$ and $m$, where $n$ refers to the number of qubits and $m$ refers to the number of parameters. We adopt $\mathcal{L}_3$ as the loss function. Training data are down-sampled, flattened, and normalized to $2^n$-dimensional state vectors. We adopt the hardware efficient ansatz [20] (Fig. 2) as the backbone of quantum circuits and use the same structure for AAE. Given a target state, a pre-trained SuperEncoder model is invoked to generate parameters and thus the circuit for QSP. While for AAE, we employ online iterations for each state. For AE, the arithmetic decomposition method in PennyLane [34, 4] is adopted. We defer more details about implementation to Appendix A. Our framework is open-source at `https://anonymous.4open.science/r/SuperEncoder-A733` with detailed instructions to reproduce our results.

### 4.2 Evaluation on Synthetic Dataset

For simplicity and without loss of generality, we focus our discussion on the results of 4-qubit QSP tasks. The outcomes for larger quantum states are detailed in Appendix B.1. The parameters of both AAE and SuperEncoder are optimized based on ideal quantum circuit simulation.

**Runtime.** The runtime and fidelity results, evaluated on the synthetic dataset, are presented in Table 3. We observe that SuperEncoder runs faster than AAE by orders of magnitudes and has a similar runtime to AE, affirming that SuperEncoder effectively overcomes the main drawback of AAE.

| | AE | | AAE | | SuperEncoder | |
|---|---|---|---|---|---|---|
| | Fidelity | Runtime | Fidelity | Runtime | Fidelity | Runtime |
| Uniform | | | 0.9996 | | 0.9731 | |
| Normal | | | 0.9992 | | 0.8201 | |
| Log-normal | | | 0.9993 | | 0.9421 | |
| Exponential | | | 0.9996 | | 0.9464 | |
| Dirichlet | | | 0.9995 | | 0.9737 | |
| Average | 1.0000 | 0.0162 s | 0.9994 | 5.0201 s | 0.9310 | 0.0397 s |

Table 3: Comparison between AE, AAE and SuperEncoder in terms of runtime and fidelity.

---

[2]`https://quantum-computing.ibm.com/`

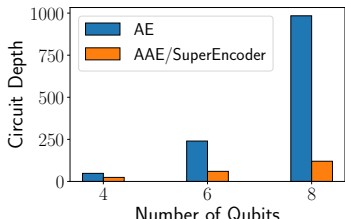 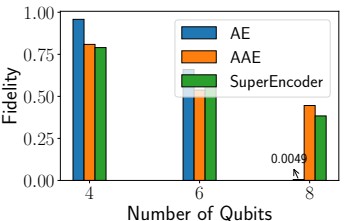

(a) Scaling of circuit depth w.r.t. # qubits.    (b) Fidelity of different QSP methods on `ibm_osaka`.

Figure 7: Comparison between AE, AAE, and SuperEncoder in terms of circuit depth and fidelity on real devices.

**Scalability.** Although AE runs fast, it exhibits poor scalability since the circuit depth grows exponentially with the number of qubits. The depth of AAE is empirically determined by increasing depth until the final fidelity does not increase, same depth is adopted for SuperEncoder. We deter the details of determining the depth of AAE/SuperEncoder to Appendix A. As shown in Fig. 7a, the depth of AE grows fast and becomes much larger than AAE/SuperEncoder, e.g., the depth of AE for a 8-qubit state is 984, whereas the depth of AAE/SuperEncoder is only 120.

**Fidelity.** From Table 3, it is evident that SuperEncoder experiences notable fidelity degradation when compared with AAE and AE. Specifically, the average fidelity of SuperEncoder is 0.9307, whereas AAE and AE achieve higher average fidelities of 0.9994 and 1.0, respectively. Note that, although AE demonstrates the highest fidelity under ideal simulation, its performance deteriorates significantly in noisy environments. Fig. 7b presents the performance of these three QSP methods on quantum states with 4, 6, and 8 qubits on the `ibm_osaka` machine. While the fidelity of AE is higher than AAE/SuperEncoder on the 4-qubit and 6-qubit states, its fidelity on the 8-qubit state is only 0.0049, becoming much lower than AAE/SuperEncoder. This decline is primarily attributed to its large circuit depth as shown in Fig. 7a.

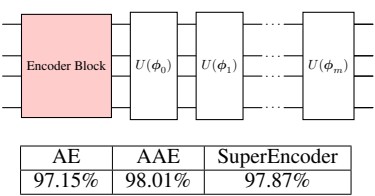

| AE | AAE | SuperEncoder |
|---|---|---|
| 97.15% | 98.01% | 97.87% |

Figure 8: Schematic of a QNN (above) and test accuracies of QSP methods on the QML task (below).

### 4.3 Application to Downstream Tasks

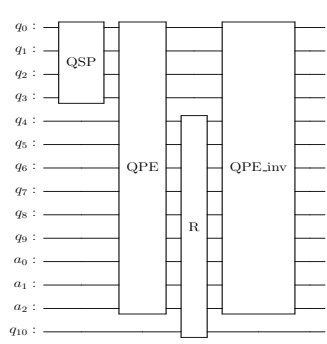

Figure 9: Schematic of HHL.

**Quantum Machine Learning.** We first apply SuperEncoder to a QML task. MNIST dataset is adopted for demonstration, we extract a sub-dataset composed on digits 3 and 6 for evaluation. The quantum circuit that implements a QNN is depicted in Fig. 8, which consists of an encoder block and $m$ entangler layers. Here the encoder block is implemented via QSP circuits, either AE, AAE, or SuperEncoder, of which the parameters are frozen during the training of QNN. The test results are shown in Fig. 8, we observe that SuperEncoder achieves similar performance with AAE and AE. The reason lies in the fact that classification tasks can be robust to noises. Consequently, approximate QSP (AAE and SuperEncoder) with a certain degree of fidelity loss is tolerable.

**HHL Algorithm.** Besides QML, quantum-enhanced linear algebra algorithms are another important set of applications that heavily rely on QSP. The most famous algorithm is the HHL algorithm [15]. The problem can be defined as, given a matrix $\mathbf{A} \in \mathbb{C}^{N \times N}$, and a vector $\mathbf{b} \in \mathbb{C}^N$, find $\mathbf{x} \in \mathbb{C}^N$ satisfying $\mathbf{Ax} = \mathbf{b}$. A typical implementation of HHL utilizes the circuit depicted in Fig. 9. The outline of HHL is as follows. (i) Apply a QSP circuit to prepare the quantum state $|\mathbf{b}\rangle$. (ii) Apply Quantum Phase Estimation [10] (QPE) to estimate the eigenvalue of $\mathbf{A}$ (iii) Apply conditioned rotation gates on ancillary qubits based on the eigenvalues (R). (iv) Apply an inverse QPE (QPE_inv) and measure the ancillary qubits to reconstruct the solution vector $\mathbf{x}$. Note that, HHL does not return the solution $\mathbf{x}$ itself, but rather an approximation of the expectation value of some operator $\mathbf{M}$ associated with $\mathbf{x}$, e.g.,

$\mathbf{x}^\dagger \mathbf{M} \mathbf{x}$. Here, we adopt an optimized version of HHL proposed by Vazquez et al. [51] for evaluation. To compare the performance between different QSP methods, we construct linear equations with fixed matrix $\mathbf{A}$ and operator $\mathbf{M}$, while we sample different vectors from our synthetic dataset as $\mathbf{b}$. Results are concluded in Table 4. Unlike QML, HHL expects precise QSP, thus we take the results from AE as the ground truth values and compare the relative error between AAE/SuperEncoder and AE. The relative error of SuperEncoder is 2.4094%, while the error of AAE is only 0.3326%.

### 4.4 Discussion and Future Work

The results of our evaluation can be concluded in two folds. (i) SuperEncoder effectively eliminates the iteration overhead of AAE, thereby becoming both fast and scalable. However, it has a notable degradation in fidelity. (ii) The impact of fidelity degradation varies across different downstream applications. For QML, the fidelity degradation is affordable as long as the prepared states are distinguishable across different classes. However, algorithms like HHL rely on precise QSP to produce the best result. In these algorithms, SuperEncoder suffers from higher error ratio than AAE.

|  | AE | AAE | SuperEncoder |
|---|---|---|---|
| $\mathbf{b}_0$ | 0.7391 | 0.7404 | 0.7355 |
| $\mathbf{b}_1$ | 0.7449 | 0.7445 | 0.7544 |
| $\mathbf{b}_2$ | 0.7492 | 0.7469 | 0.8134 |
| $\mathbf{b}_3$ | 0.7164 | 0.7099 | 0.7223 |
| $\mathbf{b}_4$ | 0.7092 | 0.7076 | 0.7155 |
| Avg err |  | 0.3326% | 2.4094% |

Table 4: Performance of different QSP methods in HHL algorithm. 'Avg err' denotes the average relative errors between AAE/SuperEncoder and AE.

Note that, the current evaluation results may not reflect the actual performance of SuperEncoder on real NISQ devices. Recent work has shown that AAE achieves significantly better fidelity than AE does [52]. This is due to the intrinsic noise awareness of AAE, as it could obtain states from noisy devices to guide updating parameters with better robustness. In essence, the proposed SuperEncoder possesses the same nature as AAE. Unfortunately, although the noise-robustness of AAE can be evaluated on a small set of test samples, it is difficult to perform noise-aware training for SuperEncoder as it requires a large dataset for pre-training. Consequently, SuperEncoder relies on huge amounts of interactions with noisy devices, thereby becoming extremely time-consuming. As a result, the effectiveness of SuperEncoder in noisy environments remains largely unexplored, which we leave for future exploration. More discussion about this perspective is in Appendix C.

## 5 Related Work

Besides QSP, there are other methods for loading classical data into quantum states. These methods can be roughly regarded as *quantum feature embedding* primarily used in QML, which maps classical data to a completely different distribution encoded in quantum states. A widely used embedding method is known as angle embedding. Li et al. have proven that this method has a concentration issue, which means that the encoded states may become indistinguishable as the circuit depth increases [26]. Lei et al. proposed an automatic design framework for efficient quantum feature embedding, resolving the issue of concentration [24]. The central idea of this framework is to search for the most efficient circuit architecture for a given classical input, which is also known as Quantum Architecture Search (QAS) [38, 30, 54]. While the application scenario of quantum feature embedding is largely limited to QML, QSP has broader usage in general quantum applications, distinguishing SuperEncoder from all aforementioned work.

## 6 Conclusion

In this work, we propose SuperEncoder, a neural network-based QSP framework. Instead of iteratively tuning the circuit parameters to approximate each quantum state, as is done in AAE, we adopt a different approach by directly learning the relationship between target quantum states and the required circuit parameters. SuperEncoder combines the scalable circuit architecture of AAE with the fast runtime of AE, as verified by a comprehensive evaluation on both synthetic dataset and downstream applications.

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

The structure of our Appendix is as follows. Appendix A provides more details of implementing SuperEncoder. Appendix B provides additional numerical results to illustrate the impact of state sizes, model architectures, and training datasets. Appendix C analyzes the estimated runtime of training SuperEncoder on real devices.

# A Implementation Details

In this section, we elaborate the missing details of SuperEncoder in the main text.

The overarching workflow of SuperEncoder is illustrated in Fig. 10. The target quantum states are input to the MLP model. Then, the MLP model generates predicted parameters based on the target states. Afterwards, the parameters are applied to the PQC to obtain the prepared quantum states. Finally, we calculate the loss based on the prepared states and target states and optimize the weights of MLP through backpropagation.

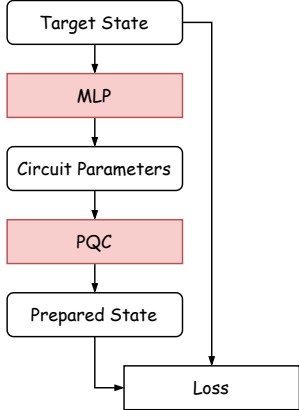

Figure 10: Detailed workflow of SuperEncoder.

The settings of MLP and PQC are as follows.

**MLP.** As listed in Table 5, we implement a two-layer MLP. Each layer consists of 512 neurons. We employ `Tanh` as the activation functions since $\boldsymbol{\theta}$ represents the *angles* of rotation gates, ranging from $-\pi$ to $\pi$.

| Linear | Input | (batch_size, $2^n$) |
|---|---|---|
| | Output | (batch_size, $512$) |
| Tanh | Input | (batch_size, $512$) |
| | Output | (batch_size, $512$) |
| Linear | Input | (batch_size, $512$) |
| | Output | (batch_size, $\dim(\boldsymbol{\theta})$) |
| Tanh | Input | (batch_size, $\dim(\boldsymbol{\theta})$) |
| | Output | (batch_size, $\dim(\boldsymbol{\theta})$) |

Table 5: MLP based SuperEncoder. $n$ refers to the number of qubits. $\boldsymbol{\theta}$ denotes the parameter vector.

**PQC.** The circuit structure is the same with the one depicted in Fig. 2, except that the number of blocks is determined dynamically through empirical examinations. Specifically, we utilize AAE to approximate a target state while increasing the number of blocks. The number of blocks is designated when the resulting state fidelity no longer increases. For example, Fig. 11 demonstrates how fidelity changes while increasing the number of blocks. As one can observe, the fidelity converges when the number of layers is larger than 8. Hence, the number of layers is set to be 8 for 4-qubit quantum states. We follow the same procedure to set the number of blocks for other state sizes. Each block has the same structure, consisting of a rotation layer and an entangler layer. Given an $n$-qubit system, a rotation layer comprises $n$ $R_y$ gates, each operating on a distinct qubit. The entangler layer is composed of two CNOT layers. The first CNOT layer applies CNOT gates to $\{(q_0, q_1), (q_2, q_3), \dots\}$, and the second CNOT layer applies CNOT gates to $\{(q_1, q_2), (q_3, q_4), \dots\}$. Hence, the depth of

a block is 3. Let $l$ be the number of blocks; then the dimension of the parameter vector is given by $\dim(\boldsymbol{\theta}) = n \times l$, and the depth of AAE/SuperEncoder is $3 \times l$. We conclude the settings of AAE/SuperEncoder used throughout this study in Table 6.

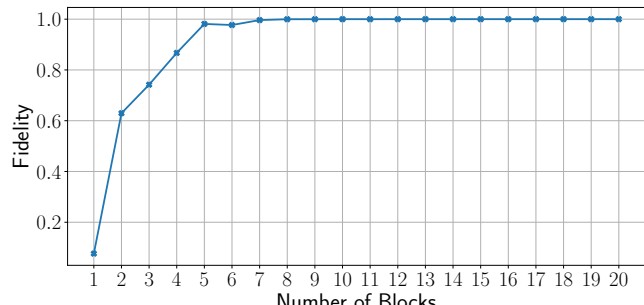

Figure 11: Fidelity vs. # blocks for 4-qubit states using AAE.

| Number of Qubits | 4 | 6 | 8 |
|---|---|---|---|
| Number of Blocks | 8 | 20 | 40 |
| Depth | 24 | 60 | 120 |

Table 6: Number of blocks and corresponding depth of AAE/SuperEncoder.

The hyperparameters for training SuperEncoder and optimizing AAE are as follows.

**Training Hyperparameters for SuperEncoder.** Throughout our experiments, the number of epochs are consistently set to be 10. For 4-qubit states, we set `bath_size` to 32, while we set it 64 for 6-qubit and 8-qubit states. We adopt Adam optimizer [22] with a learning rate of 3e-3 and a weight decay of 1e-5.

**Hyperparameters for AAE.** To optimize the parameters of AAE, we also use the Adam optimizer, with a learning rate of 1e-2 and zero weight decay. For all quantum states, we train the AAE for 100 steps.

# B  More Numerical Results

## B.1  Results on Larger Quantum States

In line with the main text, we train the SuperEncoder for 6-qubit and 8-qubit quantum states using FractalDB-60 as the training dataset. Then we evaluate the performance of SuperEncoder on the synthetic test datasets. As shown in Table 7, the average fidelity on 6-qubit and 8-qubit states are 0.8655 and 0.7624 respectively. In Appendix B.2, B.3, we discuss potential optimizations to alleviate this performance degradation.

| Dataset | $n = 4$ | $n = 6$ | $n = 8$ |
|---|---|---|---|
| Uniform | 0.9731 | 0.9254 | 0.8648 |
| Normal | 0.8201 | 0.7457 | 0.6075 |
| Log-normal | 0.9421 | 0.8575 | 0.7122 |
| Exponential | 0.9464 | 0.8757 | 0.7613 |
| Dirichlet | 0.9737 | 0.9232 | 0.8663 |
| Avg | 0.9310 | 0.8655 | 0.7624 |
| Avg-AAE | 0.9994 | 0.9964 | 0.9910 |

Table 7: Performance evaluation on larger quantum states (6-qubit and 8-qubit). The last separate row shows the results of AAE for comparison.

## B.2 Impact of Model Architecture

As a preliminary investigation, the optimal model architecture for SuperEncoder still requires further exploration. Currently, we have set the size of the hidden units at a constant 512 (Table 5). However, as the number of qubits, $n$, increases, a wider network architecture may become necessary. To showcase the impact of model width, we adjust the size to $4 \times 2^n$ for 6-qubit states and $16 \times 2^n$ for 8-qubit states, and compare their performance with the original settings, as shown in Table 8. As evident from the results, this simple adjustment significantly enhances the fidelity of SuperEncoder, suggesting that there is substantial potential to boost SuperEncoder's performance by developing a more tailored network architecture.

| Dataset | $n = 6$ | | $n = 8$ | |
|---|---|---|---|---|
| | $h = 512$ | $h = 4 \times 2^6$ | $h = 512$ | $h = 16 \times 2^8$ |
| Uniform | 0.9254 | **0.9267** | 0.8648 | **0.8821** |
| Normal | 0.7457 | **0.7580** | 0.6075 | **0.6401** |
| Log-normal | 0.8575 | **0.8608** | 0.7122 | **0.7294** |
| Exponential | **0.8757** | 0.8732 | 0.7613 | **0.7781** |
| Dirichlet | 0.9232 | **0.9261** | 0.8663 | **0.8805** |
| Avg | 0.8655 | **0.8690** | 0.7624 | **0.7820** |

Table 8: Impact of increasing network width. Here $h$ refers to the size of hidden units.

## B.3 Impact of Training Datasets

In addition to refining the model architecture, the development of a specially designed dataset for pre-training SuperEncoder is essential. Currently, the dataset utilized is FractalDB [21], which is originally designed for computer vision tasks. However, given the wide range of applications of QSP, there is a need to accommodate diverse types of classical data from various domains. Therefore, how to create a comprehensive dataset that could fully unleash the potential of SuperEncoder remains an open question. While developing a pre-trained model that performs well in all kinds of applications may be challenging, we advocate for a strategy that combines pre-training with fine-tuning for the practical deployment of SuperEncoder, similar to the approach used with foundation models in classical machine learning. To substantiate this approach, we have compiled a separate dataset that encompasses a variety of statistical distributions not limited to those utilized for evaluation (but with different settings). As demonstrated in Table 9, after fine-tuning, the performance of SuperEncoder improves by approximately 0.03.

| Dataset | Pre-training | Pre-training+Finetuning |
|---|---|---|
| Uniform | 0.9731 | **0.9909** |
| Normal | 0.8201 | **0.8879** |
| Log-normal | 0.9421 | **0.9717** |
| Exponential | 0.9464 | **0.9729** |
| Dirichlet | 0.9737 | **0.9903** |
| Avg | 0.9310 | **0.9627** |

Table 9: Fidelity improvements after fine-tuning SuperEncoder using a dataset consisting of different distributions.

## C Runtime Estimation for Training on Real Devices

Although we have theoretically analyzed the feasibility of training SuperEncoder using states from real devices (Section 3.2), its practical implementation poses significant challenges. Specifically, state-of-the-art quantum tomography techniques, such as classical shadow [55, 16], require numerous *snapshots*, each measuring a distinct observable.

To train SuperEncoder, each sample in the training dataset necessitates one classical shadow to obtain the prepared state. For instance, with the FractalDB-60 dataset, one training epoch requires 60,000 classical shadows. Our experiments on the IBM cloud platform reveal an average runtime of 3.02

seconds per circuit job excluding queuing time. Suppose the number of snapshots is 1000, then the total runtime to train SuperEncoder for 10 epochs is about 1,812,000,000 seconds[3], roughly 57 years, making the process prohibitively expensive and time-consuming.

However, quantum tomography is under active investigation, and we expect more efficient techniques to emerge for acquiring noisy quantum states from real devices. Additionally, with the advancement of quantum computing system, future systems may have tightly integrated quantum-classical hetero-geneous architectures (shorter runtime per job) while being capable of executing numerous quantum circuits in parallel (jobs within a classical shadow can execute in parallel). Hence, we anticipate the training of SuperEncoder to be feasible in the future.

---

[3] $10 \times 1000 \times 60000 \times 3.02$

