# OpenReview forum: "SuperEncoder: Towards Iteration-Free Approximate Quantum State Preparation"
_NeurIPS.cc/2024/Conference — Submitted to NeurIPS 2024_

### Official Review · Reviewer_r6Nc · 2024-07-11

**Soundness:** 3
**Presentation:** 3
**Contribution:** 3
**Rating:** 6
**Confidence:** 4

**Summary:**

This paper introduces SuperEncoder, a novel approach to Quantum State Preparation (QSP) that aims to combine the scalability of Approximate Amplitude Encoding (AAE) with the speed of traditional Amplitude Encoding (AE). SuperEncoder uses a pre-trained neural network to directly estimate the parameters of a Parameterized Quantum Circuit (PQC) for any given quantum state, eliminating the need for iterative parameter tuning during runtime. The authors explore different loss functions for training SuperEncoder, finding that state-oriented training using fidelity as a metric (L3) performs best. They evaluate SuperEncoder on synthetic datasets and downstream tasks like Quantum Machine Learning and the HHL algorithm, comparing it to AE and AAE. Results show that SuperEncoder achieves runtime similar to AE while maintaining the scalability of AAE, but with some degradation in fidelity. The impact of this fidelity loss varies across applications, being more tolerable in QML tasks than in precise algorithms like HHL.

**Strengths:**

Originality: The paper presents a novel approach to Quantum State Preparation with SuperEncoder, which innovatively combines the strengths of existing methods (AAE and AE). The idea of using a pre-trained neural network to directly estimate quantum circuit parameters is a nice solution to the QSP problem.

Quality: The research demonstrates high quality through its comprehensive experimental design. The authors explore different loss functions, provide detailed analysis of their landscapes, and evaluate the method on both synthetic datasets and real-world applications. The comparison with existing methods (AE and AAE) across multiple metrics (runtime, scalability, and fidelity) shows a rigorous approach to validation.

Clarity: The paper is well-structured and clearly written. Complex concepts are explained in an accessible manner, with helpful diagrams (like Figures 2 and 3) to illustrate key ideas.

Significance: SuperEncoder potentially represents a step towards more efficient QSP, which is crucial for many quantum algorithms.

**Weaknesses:**

1. The gradient evaluation of the loss function (e.g. Eq. 1) requires computing the derivative of the state $\rho$ with respect to model parameters. As the authors acknowledge, this could become complicated on real devices due to the enormous cost of quantum state tomography. The authors work around this by using the parameter-shift rule to compute the gradient. However, the parameter-shift rule does not scale as well as classical backpropagation with autodiff (see https://openreview.net/forum?id=HF6bnhfSqH -- I guess a citation to this work would be relevant here). This casts doubts on the whole scalability of this method.

2. Again related to scalability, the number of input neurons to the model has to be $2^n$. This again doesn't look too scalable past 20 qubits, which can already be realized experimentally.

**Questions:**

1. What are the expected compute/memory/time costs in training SuperEncoder with larger qubit numbers? Is training with more than 10 qubits feasible?

Minor:

- In ine 284: Is $m$ for the number of entangling layers the same $m$ that appears in line 242?
- In line 335: could you add a citation to the work of Li et al that you are referring to the first time it appears?

**Limitations:**

Limitations have been discussed

---

> ### Author Rebuttal · Authors · 2024-08-06
>
> We thank the reviewer for constructive feedback.
> We are elated that the reviewer found our idea a nice solution, the presentation clear, and the evaluation comprehensive.
> Following are our responses to each individual comment (which are highlighted in italics).
>
> > *The gradient evaluation of the loss function (e.g. Eq. 1) requires computing the derivative of the state $\rho$ with respect to model parameters. As the authors acknowledge, this could become complicated on real devices due to the enormous cost of quantum state tomography. The authors work around this by using the parameter-shift rule to compute the gradient. However, the parameter-shift rule does not scale as well as classical backpropagation with autodiff (see https://openreview.net/forum?id=HF6bnhfSqH -- I guess a citation to this work would be relevant here). This casts doubts on the whole scalability of this method.*
>
> We argue that parameter shift rule is not a limiting factor to the scalability of our method. The reasons are two-fold.
>
> 1. The analysis of lines 194\~216 is more a feasibility analysis. We just want to show that: *if* one wish to train SuperEncoder based on states obtained on real devices, it is possible to do so. However, it is not mandatory to train SuperEncoder on real devices; the training can be done on classical devices through noisy quantum circuit simulation.
> 2. The parameter shift rule is also not mandatory for calculating $\frac{\partial L}{\partial \theta}$. A recent study \[R0\] has introduced a hybrid method, which obtains $\hat{\rho}$ from real devices based on quantum tomography but calculates gradients based on classical backpropagation. We agree that tomography will be a bottleneck if we have to train on real device, but the tomography itself is an active research field. With more advanced tomography methods proposed, we believe the training efficiency of SuperEncoder on real devices will be significantly improved.
>
>
> We thank the reviewer for the constructive feedback and will enhance our paper with more discussions and citations regarding the overhead of gradient evaluation. We agree that such a discussion will provide a more comprehensive view for readers to understand our method.
>
> \[R0\] Wang, Hanrui, et al. "Robuststate: Boosting fidelity of quantum state preparation via noise-aware variational training." arXiv preprint arXiv:2311.16035 (2023).
>
> > *Again related to scalability, the number of input neurons to the model has to be 2^n. This again doesn't look too scalable past 20 qubits, which can already be realized experimentally.*
>
>
> We disagree that input size is a scalability issue.
>
> As stated in our paper (Sec. 2.2), the Quantum State Preparation (QSP) discussed in our paper refers to a process of **loading classical data into a quantum state**.
> Therefore, an implicit setting is that the classical data to be prepared has already been stored in classical systems, i.e., the state being prepared is within the capacity of classical storage space.
> In fact, the input to the SuperEncoder is also the input to our baselines (AE/AAE). If input size is a problem, it is a challenge for our baselines as well as the research field as a whole.
>
> Taking QML as an example, the role of QSP is loading classical image/language embeddings into quantum states. Thus the practical input size would be the same for other classical ML tasks including CV and NLP, and the number of input neurons is the same with models in these classical fields.
> If the input dimension is a problem, it will be a problem for all these classical CV/NLP models.
>
> In fact, the input size is bounded by the classical simulation power. Since quantum circuit simulation is only strictly bounded by the memory capacity. Consider an extreme case when we set the batch size to be 1, simulating a 30-qubit circuit requires a minimum of 32 GB memory (>16 GB), which can be accommodated by most modern GPUs.
> We believe this is already an enormous vector space that is capable of encoding most of classical data.
>
> > *What are the expected compute/memory/time costs in training SuperEncoder with larger qubit numbers? Is training with more than 10 qubits feasible?*
>
> As stated in the previous response, training with more than 10 qubits is absolutely feasible. We conducted experiments on a linux server with a A100 GPU that has 80 GB memory (see lines 231\~234).
> The compute/memory/time costs in training with larger qubit number are measured as follows.
>
> | Number of Qubits | Memory | Time |
> | - | - | - |
> | 10 | 960 MB | \~5 h |
> | 12 | 2520 MB | \~6 h |
> | 14 | 9722 MB | \~7.5 h |
>
> > *In line 284: Is $m$ for the number of entangling layers the same $m$ that appears in line 242?*
>
> We apologize for the confusing usage of the same symbol. These two $m$ have different meanings. $m$ in line 284 denotes the number of entangler layers in the QNN model, which is a downstream task of our propose QSP method. $m$ in line 242 is the output dimension of SuperEncoder, i.e., the classical model we use for QSP.
> We would distinguish the use of symbols in our paper to avoid confusion.
>
> > *In line 335: could you add a citation to the work of Li et al that you are referring to the first time it appears?*
>
> We apologize for the confusing citation. We will change the position of this citation to the end of line 335 for better readability.

---

> > ### Comment · Reviewer_r6Nc · 2024-08-12
> >
> > Thank you for your reply.
> >
> > Now I understand that the scope of this work does not require large qubit numbers. I would still have a follow-up question (probably more related to QML, which is the motivation to develop this technique). Let's take OpenAI's text-embedding-3-large, which in principle can be accommodated by 12 qubits. Why would we use a quantum computer (or rather QML) at this scale? Since we can simulate 12 qubits classically, what is the value of using a quantum computer here? Are you thinking that these 12 qubits are only a subset of more qubits within a single QML pipeline? Or are you rather implying that the quantum-like architecture of a layered circuit can be a useful classical model in itself?

---

> > > ### Author Response · Authors · 2024-08-13
> > >
> > > > _Are you thinking that these 12 qubits are only a subset of more qubits within a single QML pipeline?_
> > >
> > > We have found that this is truly an issue worth considering for the entire QML field.
> > > Taking ChatGPT as an example, the embedding we mentioned (3072-dim) corresponds to one token.
> > > Then question is: (a) Should we encode all tokens using one data loading quantum subroutine? (b) Or should we encode each token using a different subroutine?
> > >
> > > We have found some studies that employ the latter design (e.g., \[R0\]).
> > > In this scenario, there may be many 12-qubit data loading subroutines, if we create entanglements between all these qubits after data loading, there will be a large number of qubits (>100) and the complete system is far beyond the storage capacity of classical system.
> > >
> > > Currently, most QML research utilizes the same models as those demonstrated in our paper, which contain only one data loading block, i.e., akin to design (a). In this scenario, we may load a complete sequence of tokens using one data loading subroutine. The required number of qubits will not be very large.
> > >
> > > Which design is better is indeed an open question.
> > >
> > > However, we believe it is very safe to say: The data loading quantum subroutine will not involve very large number of qubits that is beyond the simulation power of classical computers.
> > > Therefore, we believe that the scope of our work is reasonable, and our work has practical value.
> > >
> > > \[R0\] G. Li, X. Zhao, and X. Wang, “Quantum Self-Attention Neural Networks for Text Classification,” May 11, 2022, arXiv: arXiv:2205.05625. Accessed: Jun. 03, 2022.

---

> ### Author Response · Authors · 2024-08-12
>
> Thank you for your reply and thanks for raising this interesting question.
>
> > _Why would we use a quantum computer (or rather QML) at this scale? Since we can simulate 12 qubits classically, what is the value of using a quantum computer here?_
>
> In QML, or more precisely the "quantum learning for classical data" problem, _data loading_ has long been considered as a significant obstacle \[R0\]\[R1\]\[R2\], which motivates us to conduct this study.
> Among all these previous studies of QML, the data to be loaded are all classical and thus can be accommodated by classical systems. If the qubits for loading classical data are all the qubits used in the complete quantum pipeline. Then it is true that all these QML circuits can be classically simulated.
> However, the advantage of quantum computing lies not only in its information storage capacity but also in its **information processing capabilities**.
> As discussed in the review by Biamonte, Jacob, et al.: "The input problem. Although quantum algorithms can **provide dramatic speedups for processing data**, they seldom provide advantages in reading data. This means that the cost of reading in the input can in some cases dominate the cost of quantum algorithms. Understanding this factor is an ongoing challenge."
> (The statement also highlights the importance of data loading)
>
> In other words, in QML, quantum computers are processing **the same data** as classical processors like modern GPUs. What we anticipate is a **faster processing speed** when quantum computers become more powerful.
>
> Other advantages of QML may include (1) better performance with the same amounts of parameters \[R4\]; (2) it may reach the same performance with classical model with fewer training data \[R6\]. In fact, fully understanding the advantages of QML is still an active research area.
>
> All these "quantum learning for classical data" problems assume the data to be able to be classically stored and thus do not rely on the storage capacity larger than classical system for data loading.
>
> **Or we can say, QML does not enforce large qubit numbers (at least for data loading) that are beyond the simulation capacity of classical computers.**
>
> > _Are you thinking that these 12 qubits are only a subset of more qubits within a single QML pipeline?_
>
> This is a very interesting question, it may be a possible direction to develop certain QNN architectures, where only a subset set of qubits are responsible for data loading, and there can be many additional qubits responsible for data processing.
> Then the complete circuit may be beyond the simulation capacity of classical computers.
> While we do not know many QML studies that implement circuits with such a structure, some algorithms like HHL \[R3\] do have many more ancilla qubits in addition to data loading qubits. However, we do not assume "these 12 qubits are only a subset of more qubits" for QML in our study.
>
> > _Or are you rather implying that the quantum-like architecture of a layered circuit can be a useful classical model in itself?_
>
> This is another interesting question. While we do not assume the quantum-like architecture to be a useful classical model, we are not sure whether it is entirely true that they are not useful. We do have seen some quantum-inspired designs for classical ML \[R5\].
>
> \[R0\] Biamonte, Jacob, et al. "Quantum machine learning." Nature 549.7671 (2017): 195-202.
>
> \[R1\] Caro, Matthias C., et al. "Encoding-dependent generalization bounds for parametrized quantum circuits." Quantum 5 (2021): 582.
>
> \[R2\] Li, Guangxi, et al. "Concentration of data encoding in parameterized quantum circuits." Advances in Neural Information Processing Systems 35 (2022): 19456-19469.
>
> \[R3\] A. W. Harrow, A. Hassidim, and S. Lloyd, “Quantum Algorithm for Linear Systems of Equations,” Phys. Rev. Lett., vol. 103, no. 15, p. 150502, Oct. 2009, doi: 10.1103/PhysRevLett.103.150502.
>
> \[R4\] L'Abbate, Ryan, et al. "A quantum-classical collaborative training architecture based on quantum state fidelity." IEEE Transactions on Quantum Engineering (2024).
>
> \[R5\] A. Panahi, S. Saeedi, and T. Arodz, “word2ket: Space-efficient Word Embeddings inspired by Quantum Entanglement,” Mar. 03, 2020, arXiv: arXiv:1911.04975. Accessed: Dec. 08, 2022. (ICLR'20)
>
> \[R6\] Caro, Matthias C., et al. "Generalization in quantum machine learning from few training data." Nature communications 13.1 (2022): 4919.

---

### Official Review · Reviewer_4Ck1 · 2024-07-11

**Soundness:** 2
**Presentation:** 2
**Contribution:** 1
**Rating:** 2
**Confidence:** 5

**Summary:**

In this paper, the authors propose a model, namely SuperEncoder, to solve the quantum state preparation problem. Instead of evolving the parameterized gates to generate the target quantum state, they train a model to predict the rotation parameters from the target states.

**Strengths:**

Solve the quantum state preparation problem from a new perspective.

**Weaknesses:**

1. Poor results. The results seem ok with four qubits but decrease way too fast when increasing the number of qubits. The proposed method is not comparable to previous methods.
2. It is actually impossible to use an ML model to predict the parameters. Since training the AAE ansatz is a non-convex optimization problem, finding the optimal parameter is indeed an NP-hard problem. There are infinitely many pairs of quantum states and parameters, and I wonder how the size of the training set would scale with the number of qubits.
3. The training overhead is non-negligible. If we are preparing a quantum state that is beyond the simulation power of classical devices, the evaluation methods based on state fidelity would need an enormous number of quantum circuit executions, which I suspect would not be much less than training the AAE.

**Questions:**

No questions

**Limitations:**

Naive ideas with poor experimental results.

---

> ### Author Rebuttal · Authors · 2024-08-06
>
> We thank the reviewer for their feedback. Following are our responses to each individual comment (which are highlighted in italics).
>
> > *Poor results. The results seem ok with four qubits but decrease way too fast when increasing the number of qubits. The proposed method is not comparable to previous methods.*
>
> This is an inaccurate conclusion. We compare the *results* of SuperEncoder with baselines in *three dimensions*: (1) Scalability (2) Runtime (3) Fidelity. Not all results decrease when increasing the number of qubits.
>
> - Scalability: SuperEncoder is comparable with AAE and is significantly better than AE (Fig. 7(a)).
> - Runtime: SuperEncoder is comparable with AE and is significantly better than AAE (Table 1 in the PDF attached to the global response).
> - Fidelity: SuperEncoder is worse than AE and AAE under ideal simulation (Table. 7), better than AE on real devices (Fig. 7(b)).
>
> Thus the correct conclusion is two-fold.
>
> - In terms of scalability and runtime, SuperEncoder remains comparable or better than baselines when increasing the number of qubits.
> - However, the fidelity of SuperEncoder is worse than baselines.
>
> We have acknowledged the loss of fidelity (Sec. 4.4), but we argue that this limitation does not negate the immense value and potential of this work for following reasons.
>
> 1. **We have successfully addressed the main challenges highlighted in the paper**. As emphasized in the paper (lines 51\~57), our goal is to realize a QSP method that is both *fast* and *scalable*. Here "scalable" means that the circuit depth does not grow exponentially with respect to the number of qubits. Specifically, we have addressed a significant drawback of AAE ---- the long runtime overhead introduced by iterative online optimizations. This drawback is particularly unacceptable in practical scenarios, such as using a QML model for image classification, where it is absolutely unacceptable for the data loading phase to consume the majority of the time (see lines 41\~50). Therefore, addressing this issue has significant practical value.
> 2. **Our loss in fidelity is not catastrophic.** Although SuperEncoder sacrifices fidelity, we demonstrate that it does not compromise the performance of important downstream tasks, such as QML, as shown in Figure 8. In fact, SuperEncoder remains great performance in QML when increasing the number of qubits. (see PDF attached to global response).
> 3. **Further improving the fidelity of SuperEncoder is an open challenge that will draw increased attention from our community and inspire innovative solutions.** The ultimate goal is finding a classical ML-assisted QSP method that is both fast, scalable, and with high fidelity. Achieving this goal would address a significant challenge in quantum computing and find another significant application of classical machine learning.
>
>
> > *It is actually impossible to use an ML model to predict the parameters. Since training the AAE ansatz is a non-convex optimization problem, finding the optimal parameter is indeed an NP-hard problem. There are infinitely many pairs of quantum states and parameters, and I wonder how the size of the training set would scale with the number of qubits.*
>
> This is a misunderstanding.
> - Firstly, **our framework does not involve any training of the AAE ansatz**. The training set only contains quantum states, thus there are no "pairs of quantum states and parameters". We adopt a "state-oriented" training methodology as described in lines 172\~185. Perhaps the reviewer is discussing the parameter-oriented training (lines 159\~171), this method is not adopted in our framework. In fact, we identify and address its limitation, as stated in lines 170\~171: "Consequently, required is a more effective loss function design without involving AAE." Please refer to Sec. 3.2 for more details.
> - Secondly, our empirical study **has proved that there exists some learnable mapping between target states and circuit parameters, demonstrating the possibility of using an ML model to predict the parameters**. We acknowledge that the current methodology may not be optimal, and we will continue to advance this direction in the future. Our research demonstrates that learning this mapping is non-trivial. We believe that identifying a machine learning problem that is both significantly valuable and challenging is meaningful.
>
> We thank the reviewer for the feedback and would enhance clarification in our paper.
>
>
>
> > *The training overhead is non-negligible. If we are preparing a quantum state that is beyond the simulation power of classical devices, the evaluation methods based on state fidelity would need an enormous number of quantum circuit executions, which I suspect would not be much less than training the AAE.*
>
> As stated in our paper (Sec. 2.2), the QSP discussed in our paper refers to a process of **loading classical data into a quantum state**.
> An implicit setting is that the classical data to be prepared has already been stored in classical systems, i.e., the state being prepared is within the capacity of classical storage space.
> As such, the assumption of "preparing a quantum state that is beyond the simulation power of classical devices" does not hold.
> Besides, the distinction between SuperEncoder and AAE is that: **AAE enforces training at runtime** while SuperEncoder is trained offline. For example, when using ChatGPT, the cost of model training is certainly not a concern.

---

> > ### Comment · Reviewer_4Ck1 · 2024-08-07
> >
> > Let's first find some common ground that we both agree on. The essence of this paper is to train a neural network that is used to map the target quantum state to the parameters in the ansatz. The authors claim the proposed NN can achieve comparable results with much shallower circuits compared to the previous method. (I don't think that I have any misunderstanding in the original review)
> >
> > Then comes the disagreements.
> > 1. We have infinitely many target states for any number of qubits, which means we have infinitely many pairs of target states (training data) and parameters (labels). I found it really hard to believe that such a mapping exists. Firstly, training a given quantum ansatz, as VQE does, is already a non-convex problem, and you are saying that you can "predict" all the parameters without onsite training. Secondly, we can use different ansatz (with different amounts of parameters) to achieve the same target state, and you are saying that you can map the target states to parameters in entirely different spaces. If this is possible, why do we need VQE anymore? Since you can map an arbitrary state to the ansatz parameters with your trainable neural networks.
> > 2. I've checked with other reviewers, and it seems common sense that the proposed method lacks scalability. How can you map a quantum state with $1\times 10^9$ dimension to the parameter vector with $1\times 10^9$ dimension?
> > 3. You are saying that the proposed method can achieve similar results with a shallower circuit, and I would like to point out that this is not an advantage at all. If you please try to alter the ansatz used in your experiment with extreme depth (extremely large number of parameters) and I suspect that you will find out the proposed NN is not able to map the state to such a large parameter space. The proposed method is only possible with a toy scale (including the state space and parameter space).
> >
> > I intend to keep my score.

---

> ### Author Response · Authors · 2024-08-07
>
> We thank the reviewer for the quick response, and understand that our work is counter-intuitive. Following are our response to your additional comments (highlighted in italics).
>
> > *I found it really hard to believe that such a mapping exists. Firstly, training a given quantum ansatz, as VQE does, is already a non-convex problem, and you are saying that you can "predict" all the parameters without onsite training. Secondly, we can use different ansatz (with different amounts of parameters) to achieve the same target state, and you are saying that you can map the target states to parameters in entirely different spaces. If this is possible, why do we need VQE anymore? Since you can map an arbitrary state to the ansatz parameters with your trainable neural networks.*
>
> The reviewer believe that: If the methodology of SuperEncoder is feasible, we can predict the parameters of VQE and we do not need VQE anymore. Therefore, the methodology of SuperEncoder is not feasible.
>
> We believe this is a misunderstanding.
> We argue that AAE is fundamentally different from other Hamiltonian-oriented Variational Quantum Algorithms (VQA) (e.g., VQE).
> That is, the final state that we want the system to evolve to is **NOT known** for typical VQAs, whereas the final state is known for AAE.
> Essentially, AAE belongs to QSP, but VQE does not.
> Thus, it is certainly impossible to use SuperEncoder to predict parameters for these VQAs, but this does not necessarily mean that our methodology is unfeasible.
>
> We argue that our methodology is feasible. Besides the empirical evidence provided in our paper, we would like to illustrate it from following perspective, which we have also elaborated in lines 136\~148.
>
> - In AE, i.e., the precise QSP method. For arbitrary target state, we are using exactly the same procedure to generate the required QSP circuit. That is, for any given state $|\psi\rangle$, there exists an universal mapping $f: |\psi\rangle \to U_\theta$, such that $U_\theta |0\rangle = |\psi\rangle$.
> - In this paper, we simply take one step further and are basically asking following question: Given a quantum state, is there a deterministic mapping between this state and the QSP circuit that could *approximately* prepare the state? We argue that our intuition is natural and reasonable.
>
> > *I've checked with other reviewers, and it seems common sense that the proposed method lacks scalability. How can you map a quantum state with $1\times 10^9$ dimension to the parameter vector with $1\times 10^9$ dimension?*
>
> Based on our understanding, the concerns are more about the input size and the training efficiency when the number of qubits is large.
> As we emphasized in the global response, we refer to QSP as a process of loading classical data into quantum states.
> Realistic classical data such as image/text embeddings typically do not have a exceedingly large number of dimensions.
> According to [OpenAI documentation](https://openai.com/index/new-embedding-models-and-api-updates/), its latest embedding model `text-embedding-3-large` creates embeddings with up to 3072 dimensions, which can be accommodated by 12 qubits.
>
> Moreover, it is certainly possible to use neural network to map large vectors from one space to another. The classical text-to-image tasks are great examples.
>
> > *which means we have infinitely many pairs of target states (training data) and parameters (labels)*
>
> Is "infinitely many pairs of inputs and outputs" really a problem in machine learning? As long as the mapping between inputs and outputs is learnable, it is definitely possible to construct a dataset with finite number of data points. This can be verified by all classical ML problems.
>
> > *You are saying that the proposed method can achieve similar results with a shallower circuit, and I would like to point out that this is not an advantage at all. If you please try to alter the ansatz used in your experiment with extreme depth (extremely large number of parameters) and I suspect that you will find out the proposed NN is not able to map the state to such a large parameter space. The proposed method is only possible with a toy scale (including the state space and parameter space).*
>
> We disagree with this point. Deep circuit depth is a well-known challenge in QSP, thus we definitely prefer shallower circuits.
>
>
> We sincerely thank the reviewer for the feedback and look forward to any further questions.

---

> > ### Comment · Reviewer_4Ck1 · 2024-08-07
> >
> > If the NN model can predict the parameters from the target state, then there will be an inverse model that can predict the target state from the parameters. Consider the random circuit sampling problem. We can preobtain a dataset with different circuit parameters and the final state. Under your assumption, can we train an NN model based on this dataset and predict the final state?

---

> ### Author Response · Authors · 2024-08-07
>
> > If the NN model can predict the parameters from the target state, then there will be an inverse model that can predict the target state from the parameters
>
> This assumption is almost equivalent to: For any machine learning model trained by a dataset $(x,y)$, where $x$ refers to the input and $y$ refers to the output, we can train another model using a dataset $(y,x)$, with $y$ the input and $x$ the output. We are not sure why this assumption hold.

---

> ### Author Response · Authors · 2024-08-07
>
> Dear Reviewer,
>
> Before making an ultimate judgment on the feasibility of the methodology behind SuperEncoder. We would like to humbly ask you to think about following questions.
>
> - What is the essence of AAE? It constructs a circuit with fixed structure and trainable parameters, then it iteratively update its parameters to approximate the target state. Indeed, this implies that, given an arbitrary quantum state, it is possible to utilize the same procedure to construct a QSP circuit that could approximately generate this state. However, this procedure utilizes a try-and-error methodology, its iterative optimizations at runtime becomes a bottleneck.
> - What is the essential goal of SuperEncoder? The goal is to build an AI designer to directly generate a QSP circuit that can approximately prepare an arbitrary quantum state, while minimizing online iterations to ensure efficiency. Our current work is just an initial exploration, we argue that there is significant room for further improvement. However, we may need to address some interesting but challenging research questions, which may require more people to engage in for a long time. We list some of these research questions as follows (Part of them are our ongoing work, we have to reveal these ideas for clarification).
>     - Because the procedure of precise QSP (i.e., AE) is deterministic ---- essentially arithmetic decomposition, how can we let a ML model learn from this procedure? If we can make a model understand some fundamental principles and methods of QSP, is it possible for this model to find a universal, non-iterative QSP circuit construction method while limiting circuit depth under a given approximation ratio?
>     - Currently we use a fixed circuit structure and predict its parameters. We ask, is it possible to train a model to generate different circuit structure (as well as the associated parameters) for different target states? The freedom of circuit structure, i.e., what type of gates to use and where to put them, can potentially enhance circuit expressibility. However, we may need to find an effective training methodology for this kind of models.
>
> The above dicussions may be beyond the scope of our paper. We just want to emphasize that there are many possibilities for exploration in this direction.
>
> Since we have shown the feasibility of our methodology with strong empirical evidence at the scale of 4\~8 qubits, we disagree that it is reasonable to make a conclusion that our methodology is definitely not able to be extended to larger quantum states.

---

> ### Author Response · Authors · 2024-08-08
>
> We have found a concurrent work \[R0\] that explores a similar direction with us, which has been published.
>
> In short, their method is also NN-based arbitrary QSP, but with a focus on low-level quantum control.
> Specifically, the authors proposed to "use a large number of initial and target states to train the neural network and subsequently use the well-trained network to generate the pulse sequence"
>
> Since our study started before this paper is published, we were unaware of this work. We would incorporate relevant discussions in Sec. 5.
>
> This paper serves as strong evidence of the feasibility of our work.
>
>
> \[R0\] Li, Chao-Chao, Run-Hong He, and Zhao-Ming Wang. "Enhanced quantum state preparation via stochastic predictions of neural networks." Physical Review A 108.5 (2023): 052418.

---

### Official Review · Reviewer_dsnA · 2024-07-12

**Soundness:** 3
**Presentation:** 3
**Contribution:** 3
**Rating:** 4
**Confidence:** 5

**Summary:**

The paper addresses the problem of Quantum State Preparation (QSP), which is critical for quantum computing but requires a circuit depth that scales exponentially with the number of qubits, making it impractical for large-scale problems. The authors propose SuperEncoder, a pre-trained classical neural network model designed to estimate the parameters of a Parameterized Quantum Circuit (PQC) for any given quantum state. This approach eliminates the need for iterative parameter tuning, making it a significant advancement towards iteration-free approximate QSP.

Contributions

1. Introduction of SuperEncoder, which pre-trains a classical neural network to estimate PQC parameters directly, bypassing the need for iterative updates.
2.  Provides empirical evidence that SuperEncoder significantly reduces the runtime for quantum state preparation compared to traditional methods, thus enhancing the efficiency of quantum algorithms.

**Strengths:**

See  Contributions.

**Weaknesses:**

1. [Scalability Issue]
The most significant drawback of this work is its poor scalability. Since the input to the SuperEncoder is $2^n$ dimensional, the number of qubits cannot be too high, such as exceeding 20 qubits. This limitation severely restricts the applicability of the SuperEncoder to larger quantum systems. Discussing potential strategies to overcome this drawback would greatly enhance the practical value of the SuperEncoder.

2. [Barren Plateau Problem]
Another major issue is that, even within a reasonable range of qubit numbers (e.g., 10-20), training the SuperEncoder is challenging due to the barren plateau problem. Consequently, the SuperEncoder is likely only suitable for situations involving fewer than 10 qubits. In these cases, the time difference between AAE and SuperEncoder is not as significant as one might expect, which greatly limits the potential impact of this work.

**Questions:**

See Weaknesses.

Additionally,

1. [Target Parameters Acquisition]
In the parameter-oriented training section, the process for obtaining the target parameters $\theta$ is not sufficiently clarified. If I understand correctly, these parameters are initially derived using the Variational Quantum Eigensolver (VQE) method. This approach inherently introduces errors and is prone to local minima, which could negatively impact the effectiveness of the SuperEncoder. It would be beneficial for the authors to address these issues and discuss the implications of using VQE-derived parameters.

2. [Gradient Calculation Complexity]
The gradient analysis section appears overly complex. Specifically, the calculation of gradients could be simplified by using the parameter shift rule to directly compute the gradient of $L_3$ with respect to $\theta$, rather than calculating the gradient with respect to $U$.

3. [Runtime Clarification]
In Table 3, the term "Runtime" needs clarification. It is unclear whether this refers to the training time required for the SuperEncoder or the inference time once the model is trained. Providing a clear distinction between these two would help in accurately assessing the efficiency and practicality of the proposed method.

4. [Data Distribution Scope]
The distribution characterized by the SuperEncoder seems to be specifically tailored to a particular dataset. Theoretically, the SuperEncoder should be capable of characterizing vectors across the entire space. Have the authors tested the SuperEncoder on a broader range of vectors? If so, what were the results? Addressing this question could provide valuable insights into the versatility and generalizability of the SuperEncoder.

---

> ### Author Rebuttal · Authors · 2024-08-06
>
> We thank the reviewer for constructive feedback. Following are our responses to each individual comment (which are highlighted in italics).
>
> > *[Scalability Issue] The most significant drawback of this work is its poor scalability. Since the input to the SuperEncoder is 2^n
>  dimensional, the number of qubits cannot be too high, such as exceeding 20 qubits. This limitation severely restricts the applicability of the SuperEncoder to larger quantum systems.*
>
> We disagree that scalability is a weakness. As stated in our paper (Sec. 2.2), the Quantum State Preparation (QSP) discussed in our paper refers to a process of **loading classical data into a quantum state**.
> Therefore, an implicit setting is that the classical data to be prepared has already been stored in classical systems, i.e., the state being prepared is within the capacity of classical storage space.
> In fact, the input to the SuperEncoder is also the input to our baselines (AE/AAE). If input size is a problem, it is a challenge for our baselines as well as the research field as a whole.
>
> > *[Barren Plateau Problem] Another major issue is that, even within a reasonable range of qubit numbers (e.g., 10-20), training the SuperEncoder is challenging due to the barren plateau problem. Consequently, the SuperEncoder is likely only suitable for situations involving fewer than 10 qubits.*
>
> This is a misunderstanding. "Barren Plateau" does not affect the SuperEncoder.
> The workflow of SuperEncoder is as follows.
>
> 1. Building a *classical neural network*. The input is the target state vector, the output is the parameter vector of the QSP circuit.
> 2. Training this *classical neural network*. The loss is designed to be the divergence between the state prepared by the QSP circuit and the target state.
>
> The *Barren Plateau* phenomenon occurs when **optimizing the parameters of quantum circuits** \[R0\].
> However, **parameters being optimized in SuperEncoder only include the weights of the classical neural network**.
> More specifically, the quantum circuit employed in SuperEncoder serves as a fixed tensor transformation, which maps the parameter vector generated by the classical NN model to the prepared state vector and contains no trainable parameters.
>
> > *[Target Parameters Acquisition] If I understand correctly, these parameters are initially derived using the Variational Quantum Eigensolver (VQE) method. This approach inherently introduces errors and is prone to local minima, which could negatively impact the effectiveness of the SuperEncoder.*
>
> This is a misunderstanding.
> The parameters are *not* derived using VQE. They are derived using AAE \[R1\] (our baseline). Although both AAE and VQE have the drawbacks of "inherently introduces errors" and "prone to local minima", parameter oriented training is **NOT** employed in our framework. Instead, we use *state-oriented training* without involving AAE, thereby avoiding these aforementioned drawbacks.
> In state-oriented training, we do not need to acquire target parameters. Please refer to Sec. 3.2 for more details.
>
> > *[Gradient Calculation Complexity] the calculation of gradients could be simplified by using the parameter shift rule to directly compute the gradient of $L_3$ with respect to $\theta$*
>
> We disagree with this point.
> $L_3$ is defined as $1 - \langle \psi | \hat{\rho} | \psi \rangle$ (line 196), where $|\psi\rangle$ is the target state, i.e., a constant state vector.
> $\hat{\rho}$ denotes the density matrix of the prepared state, thus we can focus on $\hat{\rho}$.
> The density matrix as a function of $\theta$ can be written as
>
> $$
> \hat{\rho} = f(U(\theta)),
> $$
>
> thus
>
> $$
> \frac{\partial \hat{\rho}}{\partial \theta} = \frac{\partial f}{\partial U} \cdot \frac{\partial U}{\partial \theta} = \frac{\partial f}{\partial U} \cdot \frac{1}{2} (U(\theta_{+}) - U(\theta_{-})),
> $$
>
> if it's possible to apply parameter shift rule to $L_3$, we have
>
> $$
> \frac{\partial f}{\partial U} \cdot \frac{1}{2} (U(\theta_{+}) - U(\theta_{-})) = \frac{1}{2} (f(U(\theta_{+})) - f(U(\theta_{-}))).
> $$
>
> This enforces $f(U(\theta_{+})) = \frac{\partial f}{\partial U} \cdot U(\theta_{+})$.
> However, the relationship between $\hat{\rho}$ and $U$ can be nonlinear due to the complexity of obtaining $\hat{\rho}$, thus $\frac{\partial L_3}{\partial \theta}$ cannot be directly calculated using the parameter shift rule.
>
>
> > *[Runtime Clarification] It is unclear whether this refers to the training time required for the SuperEncoder or the inference time once the model is trained.*
>
> Sorry about the confusion. Runtime refers to the inference time. SuperEncoder is a pre-trained model that could generate QSP circuit parameters for arbitrary target states. Training SuperEncoder is done offline and does not belong to runtime. We would clarify this definition in our paper.
>
> > *[Data Distribution Scope] The distribution characterized by the SuperEncoder seems to be specifically tailored to a particular dataset.*
>
> This is a misunderstanding. SuperEncoder is not tailored to a particular dataset. It is trained using FractalDB that contains artificial images.
> Instead of splitting FractalDB to training set and test set, we construct a test set that is independent of the training set, thereby ensuring the generalizability (Sec. 4.1).
> Specifically, the test set is composed of various distributions covering a wide range of the vector space.
> Notably, it contains state vectors sampled from uniform distribution, which can be considered as randomized states.
> The test fidelity on these states is 0.9731, affirming the generalizability of SuperEncoder.
>
>
>
> \[R0\] McClean, Jarrod R., et al. "Barren plateaus in quantum neural network training landscapes." Nature communications 9.1 (2018): 4812.
>
> \[R1\] Nakaji, Kouhei, et al. "Approximate amplitude encoding in shallow parameterized quantum circuits and its application to financial market indicators." Physical Review Research 4.2 (2022): 023136.

---

> > ### Comment · Reviewer_dsnA · 2024-08-11
> >
> > Thanks for the authors' detailed rebuttal. While I appreciate your responses, I still have a few disagreements.
> >
> > 1. Barren Plateaus: I respectfully disagree that my concern about the Barren Plateaus was a misunderstanding. I have conducted similar experiments, and while it is true that the trainable parameters are within the classical neural network (NN), the output of the NN (denoted as y) ultimately serves as the rotation angles in the quantum circuit. Applying the chain rule for gradients, we first need to compute the gradient of the loss with respect to y before computing the gradient of y with respect to the NN parameters. While the latter part is free from the barren plateau issue, the former is indeed subject to it. Therefore, the barren plateau phenomenon still poses a challenge in this context.
> >
> > 2. Data Distribution Scope: After considering the feedback from other reviewers, I tend to agree that the SuperEncoder may not fully cover the entire space. The primary contribution of this paper seems to be that the SuperEncoder can effectively handle certain particular datasets. However, the performance on these datasets may have led the authors to an optimistic view that it can scale to any qubit count, which may not be entirely accurate.
> >
> > In summary, while the SuperEncoder presents some notable contributions, I do not believe it meets the bar for acceptance at NeurIPS. Therefore, I maintain my original score.

---

> ### Author Response · Authors · 2024-08-11
>
> Thank you for your constructive feedback. We are pleased that you acknowledged the contributions of our work, and we respect your overall evaluation of this study. However, we believe that it is worthy for us to further discuss on your remained two concerns. Particularly, we lookfoward to your further response on the Barren Plateaus problem, we appreciate your time and sincerely hope to learn more during the discussion phase
>
> - **Data Distribution Scope**: In our test sets, we have randomized quantum states covering a wide range of vector space. In this context, SuperEncoder is not tailored for any particular data distribution. However, we do acknowledge that the current SuperEncoder's performance degrades with the number of qubits increases. Fully covering the entire space when increasing the number of qubits is an open challenge, we will definitely head in this direction.
>
> - **Barren Plateaus**: The gradient of loss $L$ w.r.t. weights $W$ of MLP is given by $\frac{\partial L}{\partial W} = \frac{\partial L}{\partial y} \cdot \frac{\partial y}{\partial  W}$. Here we let $y$ be the output of MLP, i.e., the parameters of quantum circuit to be consistent your previous comment. If we understand correctly, you believe that as long as this item: $\frac{\partial L}{\partial y}$ exists, it will become zero as the number of qubits increases and we will experience Barren Plateaus problem. We respectfully argue that this is questionable. As illustrated in the original Barren Plateaus paper \[R0\], Barren Plateaus occurs under the assumption that circuit is initialized to be Haar random unitary, so that variance of measurements will decrease exponentially in the number of qubits. But in our framework, we do not explicitly randomly initialize a quantum circuit since that it contains no trainable parameters. Since the parameters are the outputs from MLP, they carry the pattern of inputs and do not necessarily lead to a randomly initialized quantum circuit. In our framework, the quantum circuit acts more like a differentiable activation function. We have verified that the loss of training with 10+ qubits converges well (However it seems we could not add images here). Moreover, there have been extensive efforts on avoiding Barren Plateaus, we believe that this common concern in the entire field is not a significant drawback.
>
> \[R0\] J. R. McClean, S. Boixo, V. N. Smelyanskiy, R. Babbush, and H. Neven, “Barren plateaus in quantum neural network training landscapes,” Nat Commun, vol. 9, no. 1, p. 4812, Nov. 2018, doi: 10.1038/s41467-018-07090-4.

---

### Author Rebuttal · Authors · 2024-08-06

We thank all the reviewers for their time and comments, which have helped improve our paper.
We are pleased that the reviewers acknowledged our contributions and found our idea a novel solution.
Following are some of the main critiques; afterwards, we address each reviewer's comments individually.

- Most reviewers had concerns over the scalability of the proposed method because the input size is $2^n$. We argue that this is not a problem, because the definition of QSP in this paper is: *loading classical data into a quantum state* (lines 100\~101). The input can be any data in the classical world (e.g., embedding vectors of images, texts, or videos). These inputs are already stored in classical systems and are assumed to fit within available storage space. As such, the value of $n$ will not be exceedingly large, thus the input size cannot be a limiting factor of our method. In fact, all QSP methods discussed in our paper (i.e., AE and AAE) have the same inputs. If the input size is a problem, it would be a problem for the research field as a whole. We would incorporate a more clear problem setting in our paper to enhance clarity.
- Review dsnA and 4Ck1 misunderstood our approach. We do not rely on acquiring parameters through AAE or constructing a training dataset with pairs of states and parameters. In fact, we refer to this approach as *parameter-oriented training* in our paper, an unsuccessful approach initially explored in our study. We have identified its issue (lines 170\~171) and proposed to address it by using state-oriented training.
- Review 4Ck1 expressed concerns about the performance of SuperEncoder. We have acknowledged the degradation of fidelity and discussed this limitation (Sec. 4.4). However, we argue that this limitation does not overwhelm the immense value of our work. Firstly, the major challenges highlighted in this paper have been successfully addressed. That is, the huge overhead of iterative online optimizations in AAE. We argue that this drawback of AAE significantly hinders its practical value and addressing it would be of great significance. Secondly, our fidelity degradation is not catastrophic. Particularly, it is able to achieve excellent performance in important downstream tasks, such as QML. We include more results of QML in the attached PDF, showing that SuperEncoder can achieve performance comparable with baselines when increasing the number of qubits. Finally, we will definitely continue to investigate the solutions to further enhance the fidelity of SuperEncoder in the future. We argue that SuperEncoder initiates an interesting but also challenging machine learning problem, which is of great significance for both quantum computing and machine learning.

To summarize, we believe all concerns have been addressed. We would incorporate a part of this discussion into our paper to make it easier for readers to understand our problem settings and methodology.

---

> ### Author Response · Authors · 2024-08-09
>
> Regarding scalability concerns about the input size. We have a concrete example:
> According to [OpenAI documentation](https://openai.com/index/new-embedding-models-and-api-updates/), its latest embedding model `text-embedding-3-large` creates embeddings with up to 3072 dimensions, which can be accommodated by 12 qubits.
> For classical data of this size, our approach holds promise for efficiently achieving approximate state preparation.

---

### Decision · Program_Chairs · 2024-09-25

**Decision:**

Reject

**Comment:**

This paper studied quantum state preparation, an important problem in quantum computing. The algorithm is named SuperEncoder, a pre-trained classical neural network model designed to directly estimate the parameters of a parametrized quantum circuit for any given quantum state.

The discussion between authors and reviewers were adequate during the discussion period, and the authors attempted to address concerns raised from reviews about scalability, barren plateau, runtime, etc. However, as a summary, the paper has a main concern that the data distribution scope can be selective, whereas literature in quantum state preparation typically works for preparing an arbitrary quantum state (see for instance [1] below and references therein). In addition, there is a concurrent work [2] that was a published paper back in 2023, and the authors are obliged to clarify more about its technical novelty.

In all, although there are interesting points raised by the paper, its current version is not ready for NeurIPS 2024 from the perspective of Area Chairs, but it is encouraged that the authors can incorporate all the discussions and prepare well for the next venues.

[1] Xiao-Ming Zhang, Tongyang Li, and Xiao Yuan. "Quantum state preparation with optimal circuit depth: Implementations and applications." Physical Review Letters 129, no. 23 (2022): 230504.

[2] Chao-Chao Li, Run-Hong He, and Zhao-Ming Wang. "Enhanced quantum state preparation via stochastic predictions of neural networks." Physical Review A 108.5 (2023): 052418.